# Curcumin suppresses tumorigenesis by ferroptosis in breast cancer

**Xuelei Cao**[1ʘ], **Yao Li**[2ʘ], **Yongbin Wang**[1], **Tao Yu**[1], **Chao Zhu**[3]*, **Xuezhi Zhang**[4]*, **Jialiang Guan**[1]*

1 The Department of Emergency Internal Medicine, the Affiliated Hospital of Qingdao University, Qingdao, Shandong, China, 2 The Department of Nephrology, the Affiliated Hospital of Qingdao University, Qingdao, Shandong, China, 3 The Department of Clinical Laboratory, the Affiliated Hospital of Qingdao University, Qingdao, Shandong, China, 4 The Department of Cardiology, the Affiliated Hospital of Qingdao University, Qingdao, Shandong, China

ʘ These authors contributed equally to this work.
* 2642835772@qq.com (CZ); zxz_1204@163.com (XZ); gjlqdfy@126.com (JG)

**Data Availability Statement:** All relevant data are within the manuscript and its Supporting Information files.

**Funding:** The author(s) received no specific funding for this work.

## Abstract

Breast cancer (BC) is one of the most common malignant tumors found in females. Previous studies have demonstrated that curcumin, which is a type of polyphenol compound extracted from *Curcuma longa* underground rhizome, is able to inhibit the survival of cancer cells. However, the functional role and mechanism of curcumin in BC are still unclear. The Cell Counting Kit-8 assay was performed to examine the effects of curcumin on cell viability in the BC cell lines MDA-MB-453 and MCF-7. The levels of lipid reactive oxygen species (ROS), malondialdehyde (MDA) production, and intracellular $Fe^{2+}$ were determined to assess the effects of curcumin on cell ferroptosis. Western blot analysis was also carried out to detect the protein levels. Finally, the antitumorigenic effect of curcumin on BC was identified in a xenograft tumor model. In the present study, the results indicated that curcumin could dose-dependently suppress the viability of both MDA-MB-453 and MCF-7 cells. Further studies revealed that curcumin facilitated solute carrier family 1 member 5 (SLC1A5)-mediated ferroptosis in both MDA-MB-453 and MCF-7 cells by enhancing lipid ROS levels, lipid peroxidation end-product MDA accumulation, and intracellular $Fe^{2+}$ levels. *In vivo* experiments demonstrated that curcumin could significantly hamper tumor growth. Collectively, the results demonstrated that curcumin exhibited antitumorigenic activity in BC by promoting SLC1A5-mediated ferroptosis, which suggests its use as a potential therapeutic agent for the treatment of BC.

## Introduction

Breast cancer (BC) is one of the most frequently diagnosed cancers and the leading cause of cancer-related death among women [1]. BC accounts for 30% of all cancer cases and 14% of all cancer-related deaths among women [2]. In recent years, the incidence of BC has continued to rise, which affects human health and the quality of life, and causes a massive burden to the medical industry and economy. Due to the lack of notably early symptoms and standardized physical examinations, the majority of patients with BC are diagnosed with metastasis, which

**Competing interests:** The authors have declared that no competing interests exist.

**Abbreviations:** ACSL4, Acyl-CoA synthetase long-chain family member 4; CCK-8, Cell counting kit-8; cDNA, Complementary DNA; C-968, Compound 968; DMEM, Dulbecco's modified Eagle's medium; FBS, Fetal bovine serum; Fer-1, Ferrostatin-1; FTL, Ferritin light chain; GPX4, Glutathione peroxidase 4; MDA, Malondialdehyde; NOX1, Nicotinamide adenine dinucleotide phosphate-oxidase 1; NS, Necro-sulfonamide; PVDF, Polyvinylidene difluoride; qRT-PCR, Quantitative real-time polymerase chain reaction; ROS, Reactive oxygen species.

results in a poor prognosis [3]. Surgery, chemotherapy, and radiotherapy are most commonly used for the treatment of BC. However, chemotherapeutic drugs generally have the disadvantages of being costly and causing side effects, including emesis, nausea, alopecia, myelosuppression, and thromboembolism [4, 5]. Therefore, it is of considerable significance to identify safe, effective, and widely sourced anticancer drugs with limited side effects for the treatment of BC.

Accumulating evidence has demonstrated that ingredients extracted from Chinese herbal medicines and natural plants can be considered novel approaches to prevent and cure tumors [6]. Curcumin is the main active material that is separated from the *Curcuma longa* underground rhizome [7]. Curcumin has a widespread function in tumor prevention and treatment [8]. Curcumin exhibits antitumor effects on various cancers via the regulation of tumor-related genes and signaling pathways [9]. Recent studies have shown that curcumin exhibits antitumor effects on BC [10]. However, the functional roles and mechanisms of curcumin in BC have not been clearly elucidated.

Ferroptosis is a type of iron-dependent programmed cell death, which is different from apoptosis, necrosis and autophagy [11]. The primary mechanism underlying ferroptosis involves the action of divalent iron or lipoxygenase, which catalyzes the metabolism of unsaturated fatty acids on the cell membrane, resulting in lipid peroxidation that eventually induces cell death [12]. Ferroptosis plays an essential role in the occurrence and development of cancer [13, 14]. Moreover, recent studies have shown that the induction of cell ferroptosis may become an effective cancer treatment strategy [15]. It has been reported that danshen, a traditional Chinese medicine, improves survival of patients with BC and induces ferroptosis and apoptosis of BC cells [16]. Similarly, an additional study published in 2018 reported that certain natural compounds exerted antitumor activities via the induction of non-apoptotic programmed cell death, including ferroptosis, which provided an effective therapeutic strategy for patients with cancer [17]. The aforementioned studies indicate that ferroptosis plays an important role in the occurrence and progression of this disease. However, whether curcumin exhibits antitumor effects by regulating cell ferroptosis in BC remains unknown.

In the present study, curcumin treatment significantly inhibited BC cell viability in a dose-dependent manner. Moreover, administration of curcumin to BC cells induced ferroptosis by enhancing the levels of lipid reactive oxygen species (ROS), malondialdehyde (MDA), which is one of the most vital end-products of lipid peroxidation, and intracellular $Fe^{2+}$. Treatment of BC cells with curcumin significantly suppressed tumorigenesis by upregulating solute carrier family 1 member 5 (SLC1A5) expression, which is an essential transporter of glutamine. Based on these results, it may be concluded that therapeutic interventions mediated through the use of curcumin-induced ferroptosis can potentially provide a promising strategy for the treatment of BC.

## Materials and methods

### Cell culture

The human BC cell lines (MDA-MB-453 and MCF-7) were purchased from Shanghai Institutes for Biological Sciences Chinese Academy of Sciences (Shanghai, China). The cells were cultured in Dulbecco's modified Eagle's medium (DMEM) medium (Gibco; Thermo Fisher Scientific, USA) containing 10% fetal bovine serum (Gibco; Thermo Fisher Scientific, USA) and 2 mM L-glutamine at 37°C in an incubator with a humidified atmosphere containing 5% $CO_2$.

### Cell treatment

Curcumin (purity>98%; Sigma-Aldrich; Merck KGaA, USA) was dissolved in DMSO at a concentration of 0, 1, 2, 5, 10, 20, and 50 μM. The stock solutions were stored and diluted to

specific concentrations in cell culture medium for cell treatment. Both MDA-MB-453 and MCF-7 cells were preincubated with various inhibitors, such as ZVAD-FMK (Sigma-Aldrich; Merck KGaA, USA), ferrostatin-1 (Fer-1; Sigma-Aldrich; Merck KGaA, USA), deferoxamine (DFO; Sigma-Aldrich; Merck KGaA, USA), and necro-sulfonamide (NS; Sigma-Aldrich; Merck KGaA, USA), for 2 h. 10 nM erastin (MedChem Express, USA) was used as a control for 24 h, following treatment with different concentrations of curcumin for 48 h.

## Cell transfection

A total of $2x10^5$ cells were seeded per well and grown to 40–60% confluence. The SLC1A5 small interfering RNA (si-SLC1A5), and blank plasmid were purchased from GenePharma (Shanghai, China). The vectors were transfected into MDA-MB-453 and MCF-7 cells using Lipofectamine® 3000 kits (Invitrogen; Thermo Fisher Scientific, USA) according to the manufacturer's protocols. Following transfection, the cells were incubated for 48 h, and the transfection efficiency was determined by western blot analysis.

## Cell viability analysis

A total of 100 μl cell suspension ($5 \times 10^3$ cells) was plated in 96-well plates and incubated at 37˚C with 5% $CO_2$. The cells were grown to ~70% confluence and subsequently treated with curcumin or transfected with si-SLC1A5. Following transfection, the cells were incubated for 24 h and 10 μl Cell Counting Kit-8 (CCK-8; Beyotime Institute of Biotechnology, China) solution was added to each well. The samples were incubated for 60 min and the absorbance of each well was assessed at 450 nm using a microplate reader (Thermo Fisher Scientific, USA).

## Quantitative real-time polymerase chain reaction (qRT-PCR)

Total RNA was extracted with TRIzol® reagent (Invitrogen; Thermo Fisher Scientific, USA) according to the supplier's instructions. A total of 1 μg total RNA was reverse transcribed into single-strand complementary DNA (cDNA) using the One Step PrimeScript miRNA cDNA Synthesis Kit (TaKaRa Bio, Japan). RT-qPCR was performed in triplicate using SYBR Green PCR Master Mix (Life Technologies, USA) following the manufacturer's protocols. The relative expression levels of acyl-CoA synthetase long-chain family member 4 (ACSL4), nicotinamide adenine dinucleotide phosphate-oxidase 1 (NOX1), glutathione peroxidase 4 (GPX4), and ferritin light chain (FTL) were normalized using β-actin as reference. The values were calculated using the $2^{-\Delta\Delta Ct}$ method. The sequences of the primers used for RT-qPCR were the following: ACSL4 (SLC1A5) forward, 5'–TTTTGCGAGCTTTCCGAGTG–3' and reverse, 5'–AGCCGACAATAAAGTACGCAA–3'; NOX1 forward, 5'–TTGGGTCAACATTGGCCTGT–3' and reverse. 5'–AAGGACAGCAGATTGCGACA–3'; GPX4 forward, 5'–ATTGGTCGGCTGG ACGAG–3' and reverse, 5'–TCGATGTCCTTGGCGGAAAA–3'; FTL forward, 5'–GCCAC TTCTTCCGCGAATTG–3' and reverse, 5'–TTCATGGCGTCTGGGGTTTT–3'; Sodium-coupled neutral amino acid transporter 1 (SLC38A1) forward, 5'–AACCTCCTTAGGCA TGTCTGT–3' and reverse, 5'–GCAAAGGCGAGTCCCAAAAT–3' and β-actin forward, 5'–TCCCTGGAGAAGAGCTACGA–3' and reverse, 5'–AGCACTGTGTTGGCGTACAG–3'.

## Western blot analysis

Total protein was extracted by cell lysis using RIPA Lysis Buffer (Beyotime Institute of Biotechnology, China). The lysate was centrifuged at 12,000 rpm at 4˚C for 10 min, and the supernatant was transferred to a new tube to quantify total protein using the BCA assay. Subsequently, electrophoresis was conducted with 12% SDS gels followed by transfer onto PVDF membranes (EMD

Millipore). Following blocking with 5% (w/v) non-fat dry milk, the membranes were incubated with primary antibodies against β-actin (ab8227, 1:1,000), SLC1A5 (ab237704, 1:1,000), aspartate aminotransferase (GOT1; ab239487, 1:1,000 dilution), and glutaminase 2 (GLS2; ab113509, 1:1,000) and all antibodies were purchased from Abcam. Subsequently, the appropriate HRP-conjugated secondary antibodies (1:5,000; ProteinTech Group) were applied. The protein bands were detected using a chemiluminescence-based method (Pierce; Thermo Fisher Scientific, USA) on a Tanon 5200 Imaging system (Tanon Science & Technology Co., China). The expression levels of the proteins in each sample were normalized to those of β-actin.

## Nude mice model

Female BALB/c nude mice (age, 6–8 weeks) were purchased from Guangdong Medical Laboratory Animal Center (Foshan, China). The animal model experiments were approved by the Ethical Committee of The Affiliated Hospital of Qingdao University. MCF-7 cells ($5x10^6$) were suspended in serum-free DMEM and subsequently injected into the right posterior flanks of the mice. Following two weeks of tumor growth, 40 mice were randomly divided into the four following groups (n = 10 per group): Control group, curcumin group, curcumin + DFO group, and curcumin + NS group. The mice in the curcumin group were treated with 30 mg/kg/d curcumin (Intragastric administration). DFO and NS were administered by intraperitoneal injection at concentrations of 30 mg/kg and 30 mg/kg, respectively, three times a week following administration of curcumin. The mice in the control group were fed with 0.9% sodium chloride and 1% DMSO. The tumor growth in the mice was examined every 3 days. The mice were sacrificed by intraperitoneal injection of pentobarbital sodium (200 mg/kg) following administration of curcumin for 4 weeks, and the size of each tumor was measured. The tumor tissues were collected for subsequent experiments and the expression levels of SLC1A5 (1:500; Cell Signaling Technology) and Ki-67 (1:500; Cell Signaling Technology) were evaluated by immunohistochemical staining according to the manufacturer's instructions and previous methodologies [18].

## Iron determination assay

The intracellular ferrous iron ($Fe^{2+}$) levels were determined using the iron assay kit purchased from Abcam (cat. no. ab83366) according to the manufacturer's instructions. The experiment was repeated three times for each group.

## MDA assay

The intracellular MDA concentration in cell lysates or tissues was assessed using a lipid peroxidation assay kit (cat. no. ab118970, Abcam) according to the manufacturer's instructions. The reaction of MDA in the samples with thiobarbituric acid (TBA) resulted in the generation of a MDA-TBA adduct. The MDA-TBA adduct was quantified colorimetrically [optical density (OD) = 532 nm]. The experiments were repeated three times for each group.

## Glutamine uptake assay

BC cells were cultured in six-well plates in the glutamine-free DMEM/F-12 medium (Invitrogen; Thermo Fisher Scientific, USA). Following collection and counting, the cells were incubated with 200 nM [$^3$H]-L-glutamine (PerkinElmer; Thermo Fisher Scientific, USA) in glutamine-free DMEM/F-12 medium (Gibco; Thermo Fisher Scientific, USA) for 15 min at 37°C in the presence of curcumin or with curcumin + SLC1A5 small interfering RNA (siRNA). The cells were collected, transferred to filter paper using a 96-well plate harvester,

dried, and exposed to scintillation fluid. The counts were measured using a liquid scintillation counter (PerkinElmer; Thermo Fisher Scientific).

## Determination of lipid ROS levels

The cells ($2\times10^5$) were seeded in a 6-well plate and treated with curcumin or erastin for 24 h. All cells were cultured in DMEM medium with 5 μM BODIPY-C11 (Thermo Fisher Scientific, USA) for 45 min at room temperature. Following incubation, the cells were collected and washed twice with PBS buffer. Subsequently, the cells were resuspended in 500 μl PBS, and subsequently filtered on a 0.4 μm nylon cell strainer. Finally, they were analyzed by flow cytometry to detect the levels of ROS within the cells. The fluorescence intensities of the cells were determined using a CytoFLEX flow cytometer (Beckman Coulter, USA). Each experiment was repeated three times.

## Statistical analysis

All experiments were repeated at least three times. The data are presented as mean ± SD. Statistical analysis was performed using GraphPad Prism 8 (GraphPad Software, USA). The unpaired Student's t-test and one-way ANOVA were used to compare the means of two and three or more groups, respectively. Pairwise group comparisons were conducted using the Tukey's post hoc test following ANOVA. $P<0.05$ was considered to indicate a statistically significant difference.

# Results

## Curcumin contributes to erastin-induced ferroptosis in BC cells

Previous studies have confirmed that curcumin functions as an antineoplastic agent in various solid tumors [8]. Results of the CCK-8 assay indicated that curcumin exhibited dose-dependent inhibitory effects on both MDA-MB-453 and MCF-7 cells ($IC_{50}$-MDA-MB-453 = 19.73 μM, Fig 1A; $IC_{50}$-MCF-7 = 20.46 μM, Fig 1B). In addition, the data indicated that the corresponding volume of the DMSO solvent exhibited no cytotoxicity on MDA-MB-453 and MCF-7 cells compared with that of 20 μM curcumin (S1 Fig). Based on these findings, 20 μM curcumin was selected as the follow-up experimental dose. Moreover, the effects of this compound were examined on iron-dependent cell death of BC cells. As shown in Fig 1C and 1D, treatment with erastin, a ferroptosis activator, significantly inhibited cell viability compared with that of the control group, whereas co-treatment of curcumin and erastin strongly reduced BC cell viability. Treatment of the cells with ferrostatin-1 (Fer-1), a ferroptosis inhibitor, notably restored the inhibitory effect of curcumin or erastin on BC cell viability, whereas pretreatment of the cells with an apoptotic (ZVAD-FMK) or necroptotic inhibitor (NS) did not improve cell survival. Taken together, these results indicated that curcumin exhibited an antitumor effect on BC cells by inducing cell ferroptosis.

## Curcumin promotes lipid peroxidation and iron accumulation during activation of ferroptosis

Accumulating evidence has shown that cell ferroptosis is mainly caused by intracellular lipid peroxidation and by accumulation of lethal ROS during iron metabolism [19]. As shown in Fig 2A and 2B, treatment of the cells with erastin significantly enhanced accumulation of lipid ROS compared with the control group. Similarly, treatment of the two BC cell lines with 20 μM curcumin increased lipid ROS levels, as demonstrated by flow cytometry using the fluorescent probe C11-BODIPY. Moreover, both MDA-MB-453 and MCF-7 cells treated with erastin or curcumin indicated a rise in MDA levels, which is one of the most vital end-products of

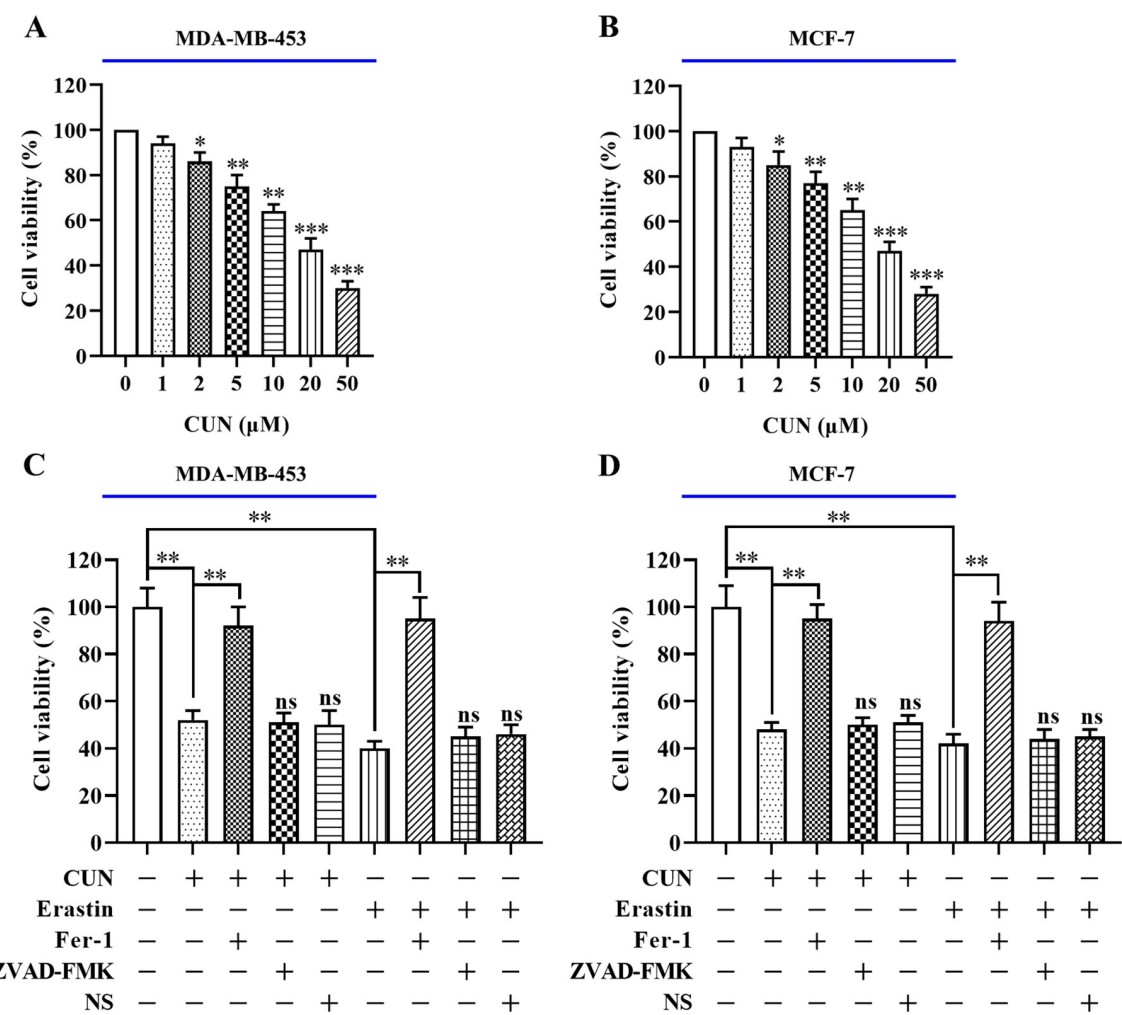

**Fig 1. Treatment with CUN upregulates erastin-induced ferroptosis in BC cells.** A and B: BC cells were treated with CUN at 0, 1, 2, 5, 10, 20 and 50 μM for 48 h. The CCK-8 assay was used to assess cell viability. C and D: BC cells were preincubated with various inhibitors, including ZVAD-FMK (50 μM), NS (50 μM), ferrostatin-1 (50 nM) or erastin (10 nM) for 2 h followed by CUN (20 μM) treatment for 48 h. The CCK-8 assay was performed to detect cell viability in both MDA-MB-453 and MCF-7 cells. Each data point is expressed as mean ± SEM of 3–5 independent tests. $^{**}P<0.01$, $^{***}P<0.001$. CUN, curcumin; BC, breast cancer; CCK-8, cell counting kit-8; NS, necro-sulfonamide; SEM, standard error of the mean.

lipid peroxidation (Fig 2C). Furthermore, higher levels of intracellular $Fe^{2+}$ were detected in the erastin or curcumin groups than those noted in the control group (Fig 2D). However, treatment of the cells with the ferroptosis inhibitor DFO caused downregulation of intracellular $Fe^{2+}$ levels (S2A Fig). It is interesting to note that treatment of the cells with curcumin or erastin significantly decreased the mRNA levels of GPX4 and ferritin light chain (FTL; Fig 2E and 2F), whereas it increased the mRNA levels of ACSL4 and NOX1 (Fig 2E and 2F). Collectively, these results indicated that curcumin significantly increased ferroptosis in BC cells.

## Increased glutamine uptake is essential for curcumin-induced ferroptosis in BC cells

Previous studies have confirmed that abnormal glutamine metabolism may contribute to ferroptosis by the accumulation of lipid peroxidation products in cancer cells (Fig 3A) [20].

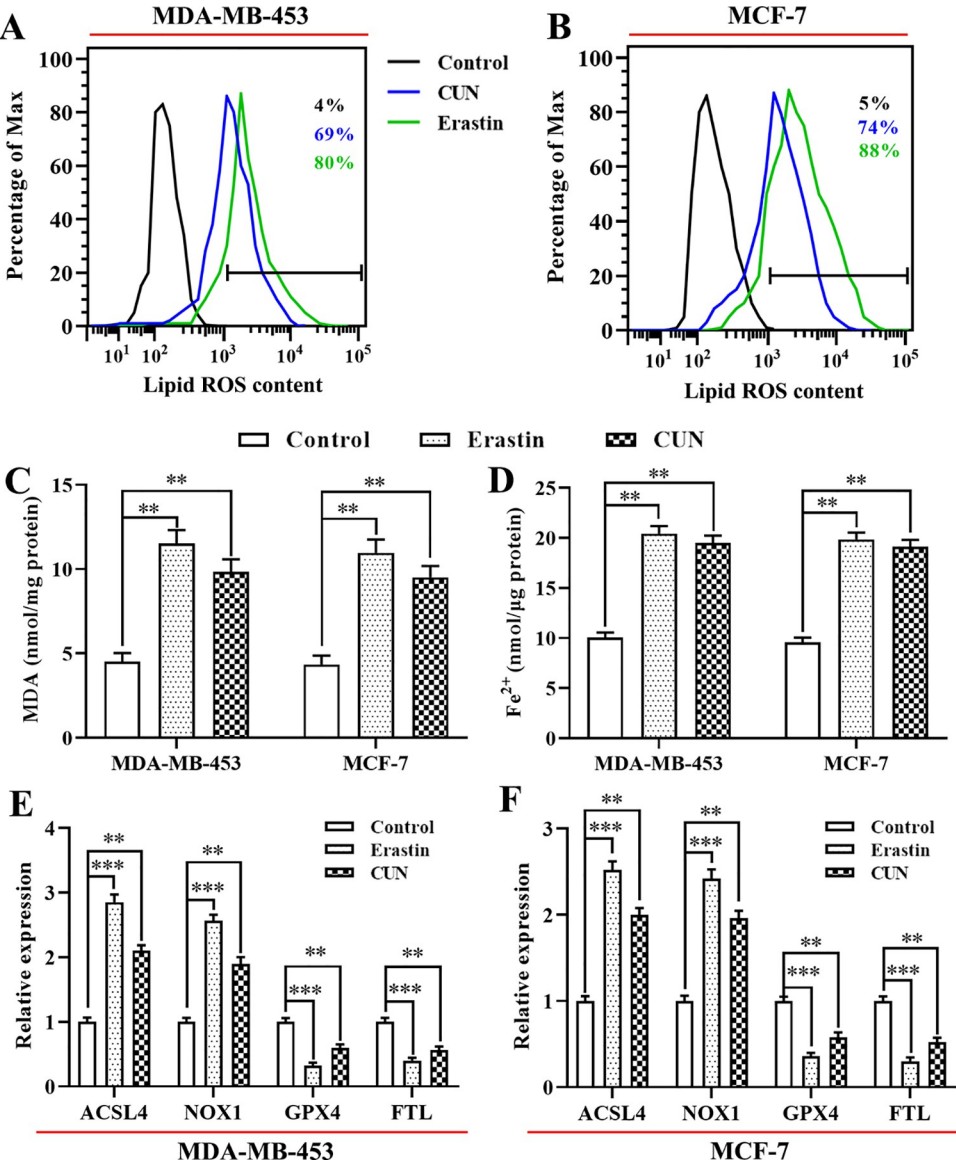

**Fig 2. CUN induces cell ferroptosis by upregulation of lipid peroxidation and iron accumulation.** A and B: The levels of lipid ROS were evaluated in MDA-MB-453 and MCF-7 cells treated with CUN for 48 h or erastin for 2 h by flow cytometry using C11-BODIPY; C: The MDA accumulation was examined by a lipid peroxidation assay kit; D: The intracellular $Fe^{2+}$ levels in the BC cell lines were measured by an iron assay kit; E and F: qRT-PCR was performed to examine the mRNA levels of genes associated with ferroptosis. Each data point is expressed as mean ± SEM of 3–5 independent tests. $^{**}P<0.01$, $^{***}P<0.001$. CUN, curcumin; ROS, reactive oxygen species; MDA, malondialdehyde; BC, breast cancer; qRT-PCR, quantitative real-time polymerase chain reaction; SEM, standard error of the mean.

Glutamine uptake is mainly dependent on specific transporters, such as cysteine-preferring transporter 2 (ASCT2; SLC1A5) and SLC38A1 [21]. As expected, curcumin significantly enhanced the mRNA levels of SLC1A5, but not those of SLC38A1 in MDA-MB-453 and MCF-7 cells (S3 Fig). Moreover, SLC1A5 has been shown to mediate uptake of glutamine, which is a conditionally essential amino acid used in rapidly proliferating tumor cells [22]. The data indicated that curcumin treatment significantly enhanced glutamine uptake (Fig 3B) in both BC cell types compared with that of the control group. Subsequently, the effects of

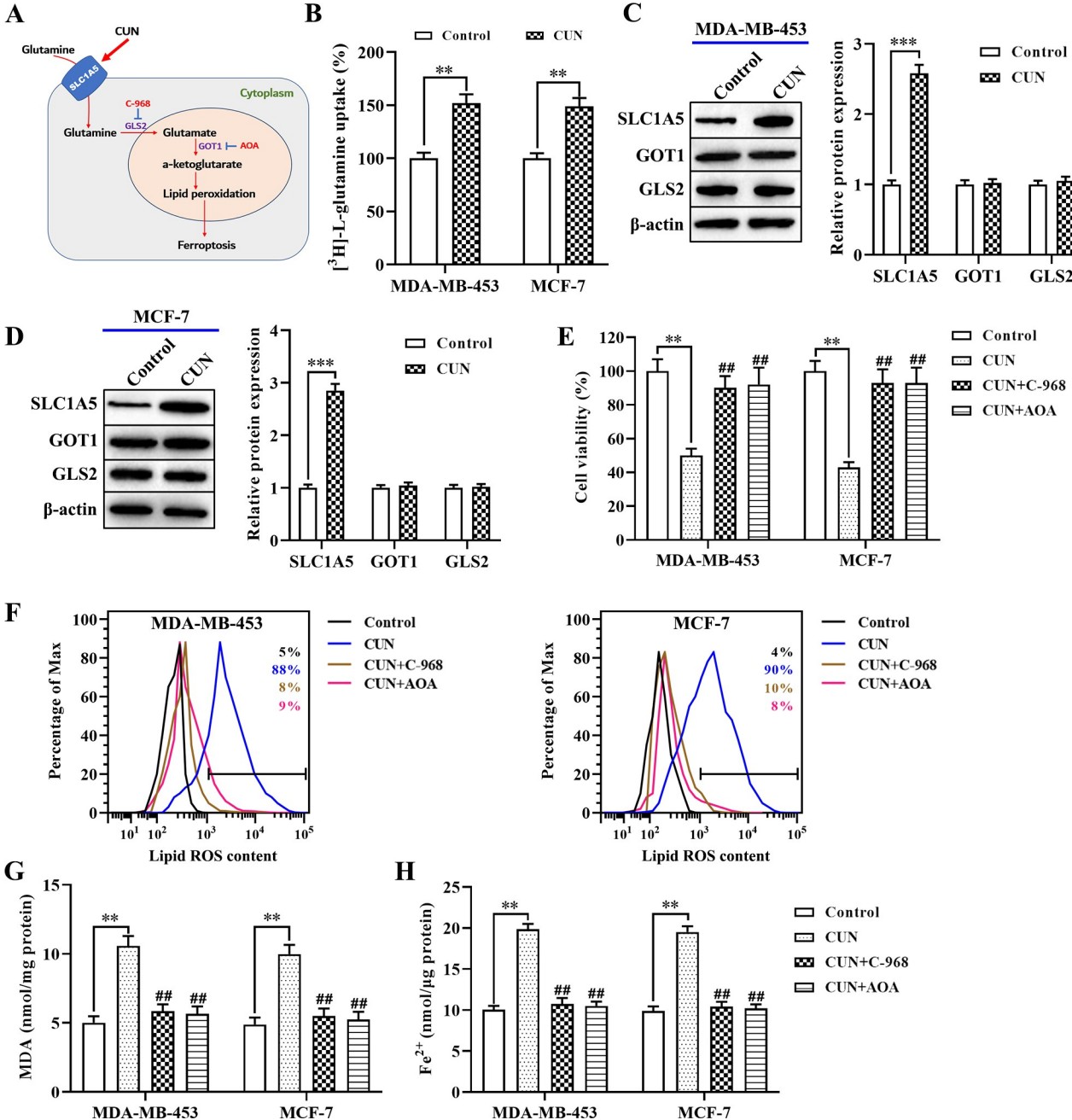

**Fig 3. Suppression of glutamine uptake is essential for CUN-induced ferroptosis in BC.** A: Schematic overview of the glutaminolysis pathway in ferroptosis; B: The [$^3$H]-$_L$-glutamine uptake was determined using a liquid scintillation counter; C and D: Western blot analysis was performed to detect the protein levels of SLC1A5, GOT1, and GLS2 in both BC cell lines; E: The CCK-8 assay was used to examine cell viability; F: Flow cytometry was employed using C11-BODIPY to detect the levels of lipid ROS in cells treated with curcumin or erastin; G: The MDA accumulation was examined by a lipid peroxidation assay kit; H: The intracellular Fe$^{2+}$ levels were measured in both BC cell lines by an iron assay kit. Each data point is expressed as the mean ± SEM of 3–5 independent tests. **P<0.01, ***P<0.001. ##P<0.01, compared with the curcumin treatment alone group. CUN, curcumin; BC, breast cancer; SLC1A5, solute carrier family 1 member 5; GOT1, aspartate aminotransferase; GLS2, glutaminase 2; CCK-8, cell counting kit-8; ROS, reactive oxygen species; MDA, malondialdehyde; SEM, standard error of the mean.

curcumin were examined on the expression of crucial glutamine metabolism genes, such as SLC1A5, GLS2, and GOT1 by western blot analysis. Both BC cell lines were treated with curcumin, which resulted in a significant increase in the protein levels of SLC1A5 compared with

those of the control group (Fig 3C and 3D). However, the protein levels of GLS2 and GOT1 remained unaltered. Moreover, both cell lines were treated with an inhibitor of GLS2 (compound 968; C-968) and GOT1 (AOA), which significantly restored the inhibitory effect of curcumin on cell viability (Fig 3E). Furthermore, curcumin induced high levels of lipid ROS (Fig 3F), MDA (Fig 3G), and intracellular $Fe^{2+}$ (Fig 3H, S2B Fig) in BC cells. These effects were alleviated by C-968 or AOA in both MDA-MB-453 and MCF-7 cells. Overall, these data suggested that curcumin induced ferroptosis by the upregulation of glutamine uptake in BC cells.

## Curcumin promotes ferroptosis by upregulating SLC1A5 expression in BC cells

To further assess the regulation of curcumin-induced ferroptosis by SLC1A5 in BC cells, SLC1A5 siRNA (si-SLC1A5) was transfected into MDA-MB-453 and MCF-7 cells in order to decrease its expression levels (Fig 4A). The CCK-8 assay indicated that curcumin treatment significantly reduced cell viability, which was ameliorated by the inhibition of SLC1A5 expression in both cell lines (Fig 4B). Moreover, knockdown of SLC1A5 reduced the effects of curcumin on glutamine uptake (Fig 4C). Similarly, the induction of lipid ROS (Fig 4D and 4E), MDA production (Fig 4F), and intracellular $Fe^{2+}$ levels (Fig 4G), which was caused by curcumin treatment in both BC cell lines was reversed by suppression of SLC1A5. Taken together, these data indicated that curcumin exhibited its antitumor effect on BC by promoting SLC1A5-mediated ferroptosis *in vitro*.

## Curcumin inhibits tumor growth of BC *in vivo*

Based on the previous *in vitro* results of curcumin on BC cells, its effects were explored on tumor growth of BC *in vivo*. As shown in Fig 5A, curcumin treatment did not affect body weight, indicating that it was safe in mice. Subsequently, the effects of curcumin were examined on tumor growth of BC *in vivo*. As shown in Fig 5B–5D, the tumor volume and tumor weight in the treatment group that received curcumin at a dose of 30 mg/kg/d were decreased compared with those of the control group. In contrast to these findings, the administration of curcumin and DFO reversed the inhibitory effect of curcumin on tumor growth, whereas the addition of NS, a necroptotic inhibitor, did not affect the efficiency of curcumin, which was consistent with the results of the *in vitro* experiments indicating that curcumin exhibited an antitumor effect on BC cells by inducing cell ferroptosis. In addition, immunohistochemical staining demonstrated that curcumin administration reduced the expression levels of Ki-67 in xenograft tissues compared with those of the control group, whereas it increased the expression levels of SLC1A5 (Fig 5E and 5F). Furthermore, curcumin administration resulted in elevated MDA production (Fig 5G) and increased iron content in tumor tissues (Fig 5H) compared with that of the control group, whereas it decreased the content of glutathione (GSH; Fig 5I). Moreover, DFO (ferroptosis inhibitor) effectively suppressed curcumin-induced ferroptosis (Fig 5B–5I), whereas NS could not change the effects of curcumin-induced cell death. In conclusion, the findings suggested that curcumin exerted its antitumor effect on BC by promoting SLC1A5-mediated ferroptosis.

## Discussion

Currently, BC exhibits the highest morbidity and mortality rates in women. Although surgery, chemotherapy, and radiotherapy are effective treatment strategies for BC, patients are still suffering from the side effects of chemoradiotherapy. Therefore, it is important to assess the mechanism of BC and seek novel strategies to reduce cancer mortality. In the present study, the results indicated that curcumin significantly suppressed cell viability and tumor growth in

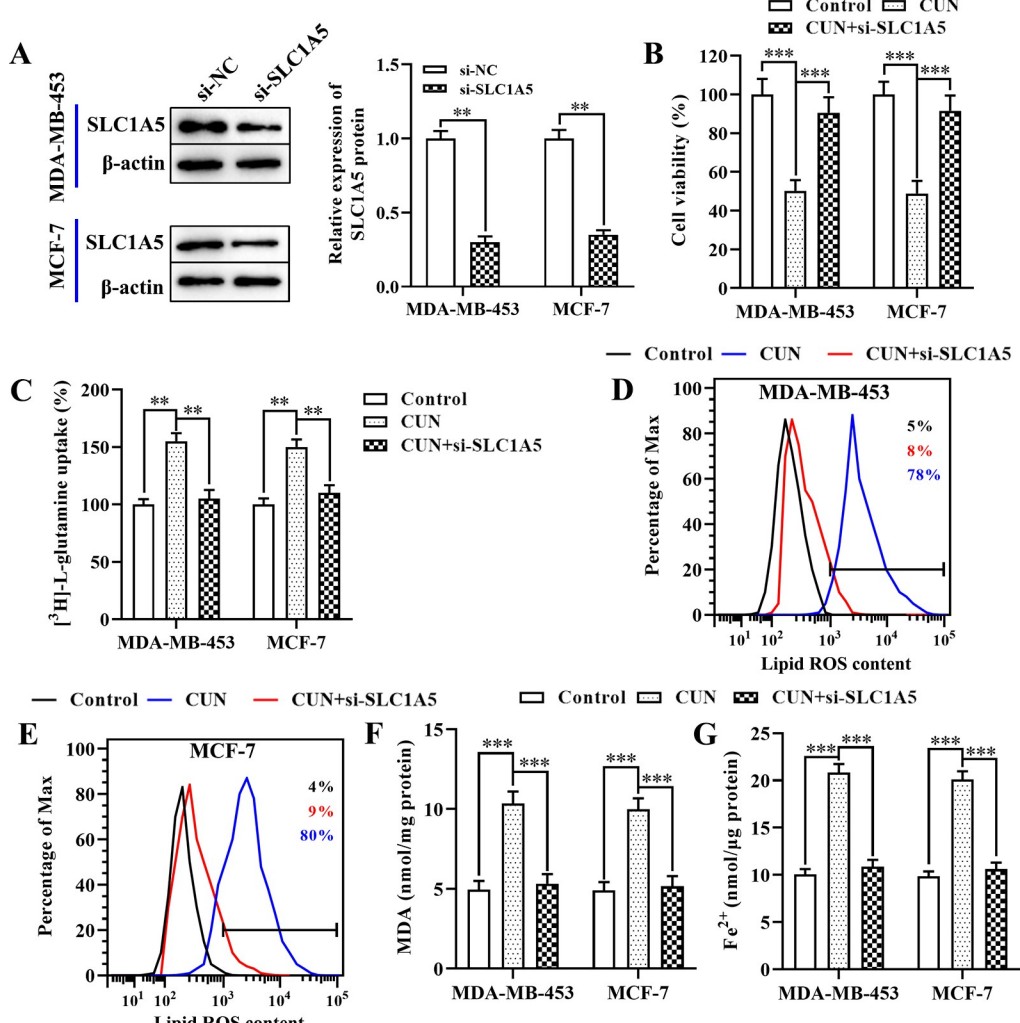

**Fig 4. Treatment with CUN facilitates ferroptosis via upregulation of SLC1A5 expression in BC.** A: Western blot analysis was performed to examine the protein levels of SLC1A5 in both BC cell lines following transfection with si-SLC1A5 or si-NC; B: The CCK-8 assay was used to examine the effects of SLC1A5 knockdown on cell viability in both BC cell lines; C: The [³H]-_L_-glutamine uptake was determined using a liquid scintillation counter; D and E: Flow cytometry was employed using C11-BODIPY to detect the levels of lipid ROS in BC cell lines treated with curcumin or erastin; F: The MDA accumulation in both BC cells was examined by a lipid peroxidation assay kit; G: The intracellular $Fe^{2+}$ levels in both BC cell lines were measured by an iron assay kit. Each data point is expressed as mean ± SEM of 3–5 independent tests. $^{**}P<0.01$. CUN, curcumin; SLC1A5, solute carrier family 1 member 5; BC, breast cancer; si-SLC1A5, SLC1A5 small interfering RNA; si-NC, NC-small interfering RNA; CCK-8, cell counting kit-8; ROS, reactive oxygen species; MDA, malondialdehyde; SEM, standard error of the mean.

BC. Moreover, BC cells treated with curcumin triggered cancer cell ferroptosis. Curcumin exhibited antitumorigenic effects on BC via the upregulation of SLC1A5-mediated ferroptosis.

Increasing evidence has confirmed that therapeutic drugs targeting cell ferroptosis can effectively interfere with cell proliferation and inhibit tumor progression [23]. Ferroptosis plays a vital role in the interaction of cancer-acquired drug resistance and immune evasion [24]. For example, inhibition of nuclear factor erythroid 2-related factor reversed cisplatin resistance by upregulating the expression levels of glutathione peroxidase 4, which is a regulator of ferroptosis, in order to induce ferroptosis in head and neck cancer [25]. Sun *et al.* reported that metallothionein-1G contributed to sorafenib resistance by suppressing

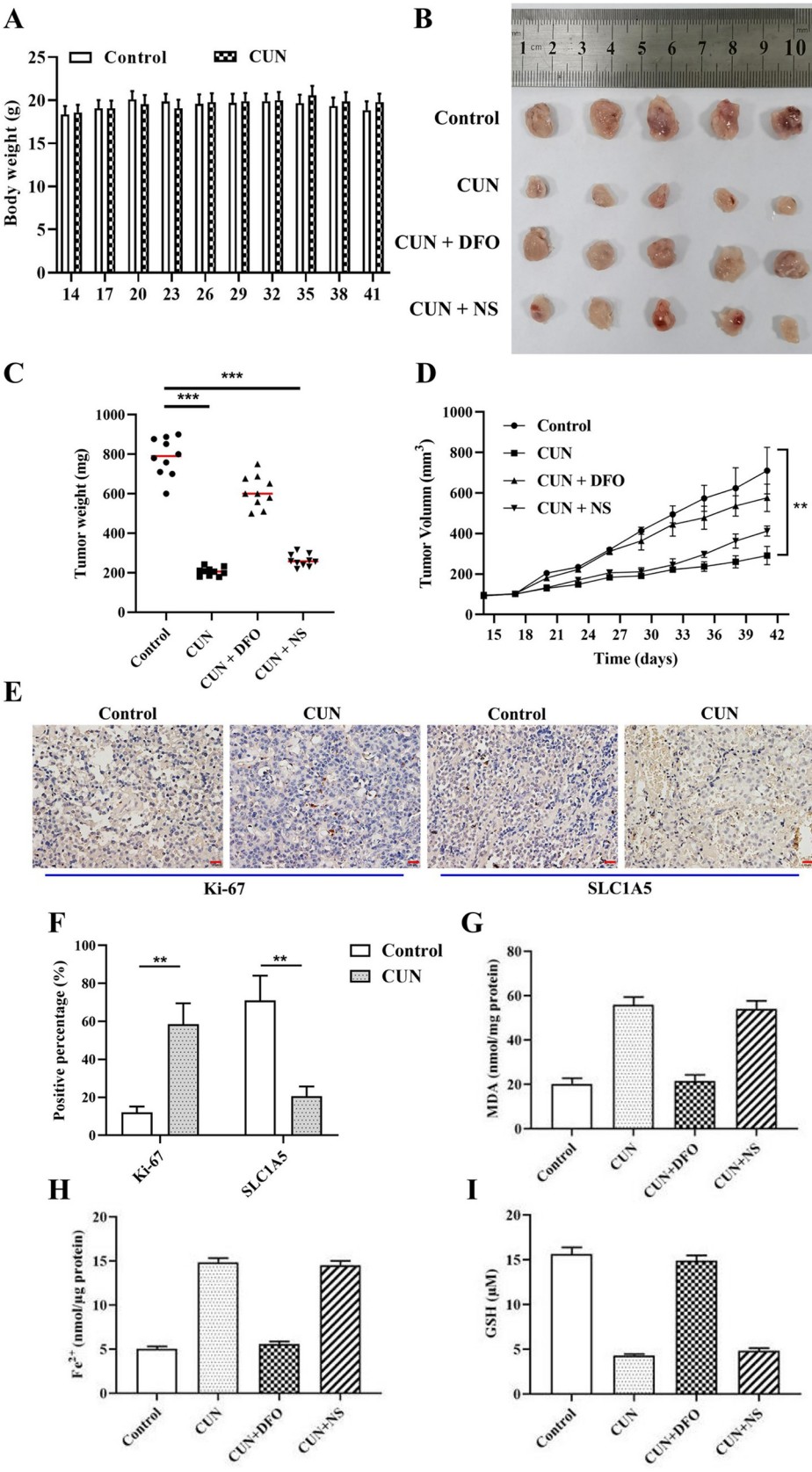

**Fig 5. CUN suppresses tumor growth *in vivo*.** MCF-7 cells were subcutaneously injected into female BALB/c nude mice (n = 10/group), followed by 30 mg/kg/d curcumin treatment for 4 weeks. DFO or NS (30 mg/kg) were administered by intraperitoneal injection three times a week following administration of curcumin. A: Measurement of the body weight of mice. B: Measurement of the gross tumor size in mice; C: The tumor volume was recorded every 3 days; D: The changes in tumor weight were measured; E, F: Immunohistochemical staining of Ki-67 and SLC1A5 expressions in tumor tissues, scale bar = 20 μm; The levels of MDA accumulation (G), iron (H) and GSH (I) were evaluated in the tumor tissues. Each data point is expressed as the mean ± SEM of three independent tests. **P<0.01. CUN, curcumin; DFO, deferoxamine; NS, necro-sulfonamide; SLC1A5, solute carrier family 1 member 5; MDA, malondialdehyde; GSH, glutathione; SEM, standard error of the mean.

ferroptosis in hepatocellular carcinoma [26]. Moreover, other studies have shown that glutamine metabolism is conducive to cell ferroptosis by enhancing the accumulation of oxidizable lipids [15, 27]. SLC1A5, which acts as an essential transporter for glutamine uptake, is associated with the progression of several tumors [28]. For example, inhibition of SLC1A5 restricted the progression of non-small cell lung cancer by decreasing glutamine consumption, cell growth, and inducing cell autophagy and apoptosis [29]. In the present study, the data indicated that inhibition of SLC1A5 promoted BC cell viability. Moreover, several studies have shown that the expression levels of SLC1A5 are higher in various solid cancers [30], while suppression of SLC1A5 activity by treatment with the glutamine transporter inhibitor L-γ-Glutamyl-p-nitroanilide or by transfecting siRNA significantly inhibits tumorigenesis [31]. In the present study, suppression of SLC1A5 inhibited ferroptosis of BC cells by reducing the levels of glutamine uptake and lipid ROS, while curcumin treatment significantly reversed this effect. Taken together, the data suggest that targeting SLC1A5 may be an effective therapeutic target for cancer treatment, particularly to target glutamine metabolism and cell biological behavior.

Accumulating evidence has confirmed that curcumin can suppress cell proliferation and induce cell apoptosis in various cancer types [32]. Moreover, curcumin exhibits anti-metastatic activity in BC cells and inhibits the proliferation of cancer stem-like cells [33], whereas it overcomes chemoresistance of cancer by suppressing the expression levels of multiple antiapoptotic proteins [34]. It is interesting to note that curcumin may act as a potential therapeutic agent and as an adjunct therapy in BC [35]. In the present study, the data indicated that treatment with curcumin notably inhibited BC cell viability and tumor growth. It is important to note that SLC1A5-mediated glutamine metabolism played an important role in regulating ferroptosis in cancer cells. As expected, the data indicated that suppression of curcumin in BC cell growth was induced by upregulation of SLC1A5-mediated ferroptosis. Similarly, several studies have demonstrated that treatment with curcumin can promote ferroptosis in non-small-cell lung cancer [36] and glioblastoma [37]. In contrast to these findings, Guerrero-Hue *et al*. demonstrated that curcumin reduced ferroptosis by inhibiting myoglobin-mediated induction of heme oxygenase-1 (HO-1) and ferritin, which contributed to cell ferroptosis [38]. A previous study by Li *et al*. confirmed that curcumin promoted BC cell ferroptosis by upregulating HO-1 levels [39]. Finally, it has been shown that HO-1 catabolizes myoglobin-derived heme to biliverdin, carbon monoxide, and iron, which is subsequently stored in ferritin [40].

In conclusion, the results of the present study indicated that curcumin inhibited cell proliferation and induced cell ferroptosis in BC. Moreover, curcumin exhibits its antitumor effect on BC by enhancing SLC1A5 expression to induce ferroptosis both *in vitro* and *in vivo*. Overall, the present study provided novel insights into the action of curcumin as an active anticancer agent, and supported the notion for its use as a potential antitumor drug for BC treatment. Moreover, the mechanism by which curcumin regulates SLC1A5-mediated ferroptosis requires further investigation.

## Supporting information

**S1 Fig. Effects of CUN on cell viability.** The CCK-8 assay was performed to assess cell viability in both BC cell lines treated with curcumin or DMSO. ***P<0.001, compared with the DMSO group. CUN, curcumin; CCK-8, cell counting kit-8; BC, breast cancer.
(TIF)

**S2 Fig. Effects of CUN on the intracellular $Fe^{2+}$ expression.** Both BC cells were preincubated with DFO (50 μM), for 2 h followed by CUN treatment for 48 h. A, B: The intracellular $Fe^{2+}$ expression in both BC cell lines was measured by an iron assay kit. **P<0.001, compared with the CUN group. CUN, curcumin; BC, breast cancer; DFO, deferoxamine.
(TIF)

**S3 Fig. Effects of curcumin on gene expression.** A and B: qRT-PCR was used to examine the mRNA expression levels of SLCA5 and SLC38A1. ###P<0.001. compared with the control group.
(TIF)

**S1 Raw images. Raw data images of western blot analysis corresponding to Figs 3 and 4.**
(PDF)

## Author Contributions

**Conceptualization:** Xuelei Cao, Jialiang Guan.

**Data curation:** Yao Li, Tao Yu.

**Formal analysis:** Tao Yu.

**Investigation:** Xuelei Cao, Yao Li.

**Methodology:** Yao Li, Yongbin Wang.

**Resources:** Chao Zhu.

**Software:** Chao Zhu.

**Supervision:** Jialiang Guan.

**Validation:** Yongbin Wang.

**Writing – original draft:** Xuelei Cao, Xuezhi Zhang, Jialiang Guan.

**Writing – review & editing:** Jialiang Guan.

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
