## [Decision Letter · Decision Letter 0]

11 Aug 2021

PONE-D-21-20659

Curcumin suppresses tumorigenesis by ferroptosis in breast cancer

PLOS ONE

Dear Dr. Guan,

Thank you for submitting your manuscript to PLOS ONE. After careful consideration, we feel that it has merit but does not fully meet PLOS ONE’s publication criteria as it currently stands. Therefore, we invite you to submit a revised version of the manuscript that addresses the points raised during the review process.

Please, pay attention, that The PLOS Data policy requires authors to make all data underlying the findings described in their manuscript fully available without restriction, with rare exception. The data should be provided as part of the manuscript or its supporting information, or deposited to a public repository. For example, in addition to summary statistics, the data points behind means, medians and variance measures should be available. If there are restrictions on publicly sharing data—e.g. participant privacy or use of data from a third party—those must be specified.

PLOS ONE requires that submissions reporting blots or gels include original, uncropped blot/gel image data as a supplement or in a public repository. This is in addition to complying with our image preparation guidelines.

We look forward to receiving your revised manuscript.

Kind regards,

Irina V. Balalaeva, PhD

Academic Editor

PLOS ONE

Journal Requirements:

Reviewers' comments:

Reviewer's Responses to Questions

**Comments to the Author**

1. Is the manuscript technically sound, and do the data support the conclusions?

Reviewer #1: Partly

Reviewer #2: Yes

Reviewer #3: Partly

2. Has the statistical analysis been performed appropriately and rigorously? 

Reviewer #1: No

Reviewer #2: Yes

Reviewer #3: I Don't Know

3. Have the authors made all data underlying the findings in their manuscript fully available?

Reviewer #1: No

Reviewer #2: Yes

Reviewer #3: Yes

4. Is the manuscript presented in an intelligible fashion and written in standard English?

Reviewer #1: No

Reviewer #2: Yes

Reviewer #3: Yes

5. Review Comments to the Author

Reviewer #1: Although the authors showed that curcumin suppressed viability of breast cancer cells and tumor growth at relatively high concentrations, it is also toxic to non-tumor cells such as mouse hippocampal HT22, PC12 and NIH-3T3 cells (Hirata et al., ACS Chem Neurosci 11, 76-85, 2020; Meng et al., PLOS ONE 9, e85570, 2014; Zhang et al., Mol Immunol 116, 191-198, 2019). Therefore, the cytotoxic effect of curcumin is not specific for breast cancer cells.

Line 61-62. This sentence is inaccurate, or misleading information.

Lin et al. analyzed a cohort of 79,335 patients with breast cancer between 2000 and 2010 and showed that danshen improves survival of breast cancer patients. They also showed that dihydroisotanshinone I (DT), a pure compound present in danshen, inhibits the growth of breast carcinoma cells, including MCF-7 and MDA-MB-231 and induces both apoptosis and ferroptosis in those cultured cells. DT-induced ferroptosis is characterized by reducing GSH content and GPX4 protein, and increasing lipid peroxidation. Therefore, there is no direct evidence that danshen prolonged the survival of patients with breast cancer by inducing ferroptosis.

Figure 2D, Figure 3H:

1. According to these results, curcumin increased ferrous ion levels in MCF-7 and MDA-MB-453 cells. However, curcumin has effective ferrous ions chelating capacity (Ak and Gulcin, Chemico-Biol Int 174, 27-37, 2008). In fact, curcumin reduces ferrous ions in cultured cells (Hirata et al., ACS Chem Neurosci 11, 76-85, 2020) and in in vitro assay (Shome et al., Biotechnology and Applied Biochemistry 68, 603-615 Biotechnology and Applied Biochemistry, 2021) at concentrations similar to those reported in this study. The authors need to examine the effect defferoxamine on ferrous ion levels as a positive control.

2. Please check whether the unit of ferrous ion is correct.

3. The relative of intracellular Fe2+ expression …: Intracellular Fe2+ level in both …

Minor points:

Figure 1: Is the concentration of erastin 10 nM?

Figure legends: The number of samples used in the experiments and statistics needs to be clarified.

Reviewer #2: In this article, the authors performed a series of in vitro and in vivo experiments to find out novel anti-tumorigenic mechanistic insight of curcumin. And they revealed that curcumin administration induced ferroptosis by enhancing the levels of lipid reactive oxygen species (ROS), malondialdehyde (MDA) of lipid peroxidation, and intracellular Fe2+ level in BC cells. Furthermore, treatment with curcumin significantly suppresses tumorigenesis in BC via upregulating SLC1A5 expression, which is an essential transporter for glutamine uptake.

As a whole, the authors performed thorough experimental plan, and their finding are quite interesting. However, the authors need further improvement of this article to grown-up to publishing-quality. The followings are the issues that the authors should be addressed.

#Because there are some other published article in which Li R, et al, revealed the anti-tumorigenic effect of curcumin by altering other ferroptosis associated gene (HO-1) in breast cancer area (Oxid Med Cell Longev. 2020 Nov 18;2020:3469840), the authors should add this article as reference and should add some discussion with regard to this article.

#As for Figure2A,2B, and 3F, please add the title of X-axis. Lipid ROS level?

#How about changing the position of Figure 5F to 5A in order to explain the less-toxic effect of curcumin to mice?

Reviewer #3: In this article, the authors show that curcumin exhibits anti-tumorigenesis activity in breast cancer cell lines by promoting SLC1A5-mediated ferroptosis (lipid ROS, lipid peroxidation end-product MDA accumulation, and intracellular Fe2+ levels). Objectives are clear, experiments are well designed, and methodology is adequate to obtain the results. The article is well written

MAJOR ISSUES

1) Which diluent is used to dissolve curcumin? It must be checked that the vehicle does not reduce cells viability.

2) Authors explain the role of ASCT2, SLC1A5 and SCL38A1 in glutamine uptake, but they only measured SLC1A5. Then they argue about the role of SLC1A5 in ferroptosis induced by curcumin but not about other receptors. Other receptors should be measured to confirm SLC1A5 implication.

3) Inmunohistochemistry staining should be quantified; a representative image is not enough to conclude Ki-67 and SLC1A5 increase.

4) To confirm the role of ferroptosis in vivo, authors should perform the same experiment but treating mice with ferrostatin and other cell death inhibitors, such as ZVAD and NS. In addition, authors should measure other ferroptosis markers than just MDA in the tumor to conclude the role of this pathway.

5) In the discussion sections, authors report that curcumin promotes ferroptosis. However, there are contradictory data in the bibliography about this issue, see reference 38 (Guerrero-Hue et al.). Authors should discuss about the possible dual role of curcumin on ferroptosis.

6) Authors should add in the figure legend the time in which each experiment was performed.

6. PLOS authors have the option to publish the peer review history of their article (what does this mean?). If published, this will include your full peer review and any attached files.

Reviewer #1: No

Reviewer #2: No

Reviewer #3: **Yes: **Juan Antonio Moreno

---

## [Author Response · Author response to Decision Letter 0]

29 Sep 2021

RE: Manuscript ID: PONE-D-21-20659

Dear Dr. Balalaeva,

We are indeed very grateful to the careful and thoughtful comments of three reviewers and your suggestions on our manuscript entitled “Curcumin suppresses tumorigenesis by ferroptosis in breast cancer” by Cao X et al. Based on the suggestions from you and the reviewers, we have carried out some additional experiments and revised the manuscript. In this letter, we have listed our responses to the specific comments/questions raised by each reviewer, and incorporated all necessary changes in the revision.

Editor:

Response: We revised our manuscript to meet PLOS ONE's style requirements.

Response: We updated the Data Availability statement in this revision by stating that “All relevant data are within the paper and its Supporting information files”.

Response: Raw images of Western blot from Figs 3 and 4 were provided as S1 Raw images in this revision.

Response: An ORCID iD for the corresponding author was updated in this submission.

Again, we really appreciate the reviewer’s careful reading and suggestions on our manuscript, and thank you very much for your consideration of our paper for publication in Plos one.

Sincerely yours,

Jialiang Guan,

The Department of Emergency Internal Medicine, 

The Affiliated Hospital of Qingdao University,

E-mail: gjlqdfy@126.com

Reviewer #1:

Question #1: Although the authors showed that curcumin suppressed viability of breast cancer cells and tumor growth at relatively high concentrations, it is also toxic to non-tumor cells such as mouse hippocampal HT22, PC12 and NIH-3T3 cells (Hirata et al., ACS Chem Neurosci 11, 76-85, 2020; Meng et al., PLOS ONE 9, e85570, 2014; Zhang et al., Mol Immunol 116, 191-198, 2019). Therefore, the cytotoxic effect of curcumin is not specific for breast cancer cells.

Response #1: Thanks for your nice comments on our article. Recently, several studies have demonstrated that curcumin had anticancer and chemoprevention effects on breast cancer through inhibiting cancer proliferation [1] and tumor metastasis [2], as well as enhancing the sensitivity of breast cancer cells to chemotherapeutic drugs [3-5]. Moreover, our data showed that 2-50 μM curcumin significantly reduced the viabilities of breast cancer cell lines (MDA-MB-453 and MCF-7). Therefore, the above results indicated the curcumin serves as an adjunct therapy in breast cancer.

Question #2: Line 61-62. This sentence is inaccurate, or misleading information.

Response #2: We rewrote this sentence as “It was reported that danshen, a traditional Chinese medicine, improved survival of patients with breast cancer and induced ferroptosis and apoptosis of breast cancer cells”. (Line 62-63, page 3 in this revision.)

Question #3: Lin et al. analyzed a cohort of 79,335 patients with breast cancer between 2000 and 2010 and showed that danshen improves survival of breast cancer patients. They also showed that dihydroisotanshinone I (DT), a pure compound present in danshen, inhibits the growth of breast carcinoma cells, including MCF-7 and MDA-MB-231 and induces both apoptosis and ferroptosis in those cultured cells. DT-induced ferroptosis is characterized by reducing GSH content and GPX4 protein, and increasing lipid peroxidation. Therefore, there is no direct evidence that danshen prolonged the survival of patients with breast cancer by inducing ferroptosis.

Response #3: In this study, the results showed that the use of Danshen ≥84 g was highly associated with decreased mortality (the adjusted HR of Danshen ≥84 g users was 0.54 [95% CI, 0.46–0.63] (p <0.001). Moreover, the use of Danshen for >28 days remained highly associated with decreased mortality (the adjusted HR of Danshen users for >28 days was 0.55 [95% CI, 0.49–0.62] (p <0.001). Thus, these data demonstrate the protective effects of a higher dose or longer use of Danshen for patients with breast cancer.

Question #4: Figure 2D, Figure 3H:

(1). According to these results, curcumin increased ferrous ion levels in MCF-7 and MDA-MB-453 cells. However, curcumin has effective ferrous ions chelating capacity (Ak and Gulcin, Chemico-Biol Int 174, 27-37, 2008). In fact, curcumin reduces ferrous ions in cultured cells (Hirata et al., ACS Chem Neurosci 11, 76-85, 2020) and in in vitro assay (Shome et al., Biotechnology and Applied Biochemistry 68, 603-615 Biotechnology and Applied Biochemistry, 2021) at concentrations similar to those reported in this study. The authors need to examine the effect defferoxamine on ferrous ion levels as a positive control.

Response #4(1): As suggested, we examined the effect deferoxamine (DFO) on the curcumin-induced increase in intracellular iron, and the results as showed in supplementary Fig. 2. We added this information in line 210-212, page 9 in this revision.

(2). Please check whether the unit of ferrous ion is correct.

Response #4(2): As suggested, we checked the unit of ferrous and revised this error in Figs 2 and 3 in this revision.

(3). The relative of intracellular Fe2+ expression …: Intracellular Fe2+ level in both …

Response #4(3): We corrected the “The relative of intracellular Fe2+ expression” into “The intracellular Fe2+ level”. (Line 483 and 493, page 17 in this revision)

Minor points:

Question #5: Figure 1: Is the concentration of erastin 10 nM?

Response #5: Thanks for your carefully checking. In this study, the concentration of curcumin in both breast cancer cells was 10 nM.

Question #6: Figure legends: The number of samples used in the experiments and statistics needs to be clarified.

Response #6: We have added the detailed information for the samples used in the figure legend in this revision. (Line 477, 485, 494, 504 and 513, page 17-18 in this revision)

Reviewer #2

In this article, the authors performed a series of in vitro and in vivo experiments to find out novel anti-tumorigenic mechanistic insight of curcumin. And they revealed that curcumin administration induced ferroptosis by enhancing the levels of lipid reactive oxygen species (ROS), malondialdehyde (MDA) of lipid peroxidation, and intracellular Fe2+ level in BC cells. Furthermore, treatment with curcumin significantly suppresses tumorigenesis in BC via upregulating SLC1A5 expression, which is an essential transporter for glutamine uptake.

As a whole, the authors performed thorough experimental plan, and their finding are quite interesting. However, the authors need further improvement of this article to grown-up to publishing-quality. The followings are the issues that the authors should be addressed.

Question #1: Because there are some other published articles in which Li R, et al, revealed the anti-tumorigenic effect of curcumin by altering other ferroptosis associated gene (HO-1) in breast cancer area (Oxid Med Cell Longev. 2020 Nov 18; 2020:3469840), the authors should add this article as reference and should add some discussion with regard to this article.

Response #1: As suggested, we cited this reference as Ref.39 and added some discussion in line 304-308, page 12 in this revision.

Question #2: As for Figure2A,2B, and 3F, please add the title of X-axis. Lipid ROS level?

Response #2: As suggested, we added the title of X-axis as “Lipid ROS content” in Figure 2A, 2B and 3F.

Question #3: How about changing the position of Figure 5F to 5A in order to explain the less-toxic effect of curcumin to mice?

Response #3: As suggested, we exchanged the position of Figure 5F and 5A.

Reviewer #3

In this article, the authors show that curcumin exhibits anti-tumorigenesis activity in breast cancer cell lines by promoting SLC1A5-mediated ferroptosis (lipid ROS, lipid peroxidation end-product MDA accumulation, and intracellular Fe2+ levels). Objectives are clear, experiments are well designed, and methodology is adequate to obtain the results. The article is well written.

Major issues

Question #1: Which diluent is used to dissolve curcumin? It must be checked that the vehicle does not reduce cells viability.

Response #1: As suggested, we provided the dilution method of curcumin and test the cell viability of DMSO solvent. As shown in supplementary Fig. 1, the corresponding volume of DMSO had no cytotoxicity on MDA-MB-453 and MCF-7 cells (supplementary Fig. 1). (Line 86-88, page 4 and line 189-191, page 8 in this revision.)

Question #2: Authors explain the role of ASCT2, SLC1A5 and SLC38A1 in glutamine uptake, but they only measured SLC1A5. Then they argue about the role of SLC1A5 in ferroptosis induced by curcumin but not about other receptors. Other receptors should be measured to confirm SLC1A5 implication.

Response #2: ASCT2 is encoded by SLC1A5 that was evaluated in this original manuscript. As suggested, we further confirmed the mRNA expression of SLC38A1 in MDA-MB-453 and MCF-7 cells after treated with or without curcumin. As shown in supplementary Fig. 3, curcumin significantly enhanced the expression SLC1A5, but had no obvious effect on the expression of SLC38A1. We added this information in line 221-222, page 9 in this revision.

Question #3: Immunohistochemistry staining should be quantified; a representative image is not enough to conclude Ki-67 and SLC1A5 increase.

Response #3: As suggested, we provided the quantified results of Immunohistochemistry staining as Figure 5F in this revision.

Question #4: To confirm the role of ferroptosis in vivo, authors should perform the same experiment but treating mice with ferrostatin and other cell death inhibitors, such as ZVAD and NS. In addition, authors should measure other ferroptosis markers than just MDA in the tumor to conclude the role of this pathway.

Response #4: As suggested, to confirm that curcumin inhibited tumor growth in vivo by ferroptosis, we treated mice with curcumin, curcumin + DFO and curcumin + NS. As shown in Figure 5B-D in this revision, the administration of curcumin + DFO obviously reversed the inhibitory effect of curcumin on tumor growth, whereas the addition of NS, did not impact the efficiency of curcumin, which was consistent with the results of in vitro experiments that curcumin showed an antitumor effect on BC cells by inducing cell ferroptosis. (Line 248-255, page 10 in this revision.)

In addition, we supplemented the results of other ferroptosis markers such as GSH and iron in tumor tissues of mice and the results were shown in Figure 5G-I. (Line 259-262, page 10 in this revision.)

Question #5: In the discussion sections, authors report that curcumin promotes ferroptosis. However, there are contradictory data in the bibliography about this issue, see reference 38 (Guerrero-Hue et al.). Authors should discuss about the possible dual role of curcumin on ferroptosis.

Response #5: Thanks for your comments, we have supplemented the possible dual role of curcumin on ferroptosis in the Discussion section. (Line 304-308, page 12 in this revision.)

Question #6: Authors should add in the figure legend the time in which each experiment was performed.

Response #6: As suggested, we added the treatment time in which each experiment in the figure legend. (Line 474, 476, 482 and 508, page 17-18 in this revision.)

References in this letter:

1. Mohammed F, Rashid-Doubell F, Taha S, Cassidy S, Fredericks S. Effects of curcumin complexes on MDAMB231 breast cancer cell proliferation. Int J Oncol. 2020;57(2):445-55. Epub 2020/07/07. doi: 10.3892/ijo.2020.5065. PubMed PMID: 32626932; PubMed Central PMCID: PMCPMC7307592.

2. Hu CX, Li MJ, Guo TT, Wang SX, Huang WP, Yang K, et al. Anti-metastasis activity of curcumin against breast cancer via the inhibition of stem cell-like properties and EMT. Phytomedicine. 2019;58. doi: ARTN 152740

10.1016/j.phymed.2018.11.001. PubMed PMID: WOS:000473047100001.

3. Xiong K, Zhang Y, Wen Q, Luo J, Lu Y, Wu Z, et al. Co-delivery of paclitaxel and curcumin by biodegradable polymeric nanoparticles for breast cancer chemotherapy. Int J Pharm. 2020;589:119875. Epub 2020/09/13. doi: 10.1016/j.ijpharm.2020.119875. PubMed PMID: 32919003.

4. Kalluru H, Kondaveeti SS, Telapolu S, Kalachaveedu M. Turmeric supplementation improves the quality of life and hematological parameters in breast cancer patients on paclitaxel chemotherapy: A case series. Complement Ther Clin Pract. 2020;41:101247. Epub 2020/10/26. doi: 10.1016/j.ctcp.2020.101247. PubMed PMID: 33099272.

5. Wen C, Fu L, Huang J, Dai Y, Wang B, Xu G, et al. Curcumin reverses doxorubicin resistance via inhibition the efflux function of ABCB4 in doxorubicinresistant breast cancer cells. Mol Med Rep. 2019;19(6):5162-8. Epub 2019/05/07. doi: 10.3892/mmr.2019.10180. PubMed PMID: 31059026; PubMed Central PMCID: PMCPMC6522915.

---

## [Decision Letter · Decision Letter 1]

19 Oct 2021

PONE-D-21-20659R1Curcumin suppresses tumorigenesis by ferroptosis in breast cancerPLOS ONE

Dear Dr. Guan,

Thank you for submitting your manuscript to PLOS ONE. After careful consideration, we feel that it has merit but does not fully meet PLOS ONE’s publication criteria as it currently stands. Therefore, we invite you to submit a revised version of the manuscript that addresses the points raised during the review process.

ACADEMIC EDITOR: Please, check the concentrations of the compounds in the experiments, especially those pointed at by the Reviewers.

 Please submit your revised manuscript by Dec 03 2021 11:59PM. If you will need more time than this to complete your revisions, please reply to this message or contact the journal office at plosone@plos.org. Please include the following items when submitting your revised manuscript:A rebuttal letter that responds to each point raised by the academic editor and reviewer(s). You should upload this letter as a separate file labeled 'Response to Reviewers'.A marked-up copy of your manuscript that highlights changes made to the original version. You should upload this as a separate file labeled 'Revised Manuscript with Track Changes'.An unmarked version of your revised paper without tracked changes. You should upload this as a separate file labeled 'Manuscript'.If applicable, we recommend that you deposit your laboratory protocols in protocols.io to enhance the reproducibility of your results. Protocols.io assigns your protocol its own identifier (DOI) so that it can be cited independently in the future. For instructions see: https://journals.plos.org/plosone/s/submission-guidelines#loc-laboratory-protocols. Additionally, PLOS ONE offers an option for publishing peer-reviewed Lab Protocol articles, which describe protocols hosted on protocols.io. Read more information on sharing protocols at https://plos.org/protocols?utm_medium=editorial-email&utm_source=authorletters&utm_campaign=protocols.

We look forward to receiving your revised manuscript.

Kind regards,

Irina V. Balalaeva, PhD

Academic Editor

PLOS ONE

Journal Requirements:

Reviewers' comments:

Reviewer's Responses to Questions

**Comments to the Author**

1. If the authors have adequately addressed your comments raised in a previous round of review and you feel that this manuscript is now acceptable for publication, you may indicate that here to bypass the “Comments to the Author” section, enter your conflict of interest statement in the “Confidential to Editor” section, and submit your "Accept" recommendation.

Reviewer #1: (No Response)

Reviewer #2: All comments have been addressed

2. Is the manuscript technically sound, and do the data support the conclusions?

Reviewer #1: No

Reviewer #2: Yes

3. Has the statistical analysis been performed appropriately and rigorously? 

Reviewer #1: Yes

Reviewer #2: Yes

4. Have the authors made all data underlying the findings in their manuscript fully available?

Reviewer #1: Yes

Reviewer #2: Yes

5. Is the manuscript presented in an intelligible fashion and written in standard English?

Reviewer #1: Yes

Reviewer #2: Yes

6. Review Comments to the Author

Reviewer #1: The authors have not adequately addressed my concerns.

Question #4/Response #4(1)

(1) The authors did not discuss the chelating ability of curcumin and ignored the related references (Ak and Guicin, 2008; Hirata et al., 2020, Shome et al., 2021). The chelating ability of curcumin is opposite action the authors reported in this study and therefore the authors should discuss the discrepancy.

The authors examined the effect of deferoxamine (DFO) on the curcumin-induced increase in intracellular Fe2+ (Figure S2). DFO only affected curcumin-induced increase in intracellular ferrous ions. The results is strange. Why DFO did not decrease the control level of ferrous ions?

Question #4/Response #4(2)

(2) The authors correct the unit of ferrous ion μg/μg protein to nmol/μg protein. According to the abcam’s protocol (ab83366, abcam) the authors used, rat liver lysate contains approximately 0.1 nmoles Fe2+/mg tissue, which is approximately 10,000 times less compared the amount of Fe2+ the authors reported (10 µmoles/µg protein).

Question #5/Response #5

Erastin at 10 nM cannot cause ferroptotic cell death in cultured cells. Most cancer cells are required 5-10 µM erastin to cause ferroptosis.

Reviewer #2: The authors have satisfactorily addressed the issues raised by reviewers and have improved the manuscript. Therefore the revised article has reached to sufficient quality.

7. PLOS authors have the option to publish the peer review history of their article (what does this mean?). If published, this will include your full peer review and any attached files.

Reviewer #1: No

Reviewer #2: No

---

## [Author Response · Author response to Decision Letter 1]

29 Oct 2021

Response to reviewer

Reviewer #1:

Question #4/Response #4(1): The authors did not discuss the chelating ability of curcumin and ignored the related references (Ak and Guicin, 2008; Hirata et al., 2020, Shome et al., 2021). The chelating ability of curcumin is opposite action the authors reported in this study and therefore the authors should discuss the discrepancy.

The authors examined the effect of deferoxamine (DFO) on the curcumin-induced increase in intracellular Fe2+ (Figure S2). DFO only affected curcumin-induced increase in intracellular ferrous ions. The results are strange. Why DFO did not decrease the control level of ferrous ions?

Response: We felt great thanks for your professional review work on our article. As suggested, we added the related references (Ak and Guicin, 2008; Hirata et al., 2020, Shome et al., 2021) as Ref. 41-43 and discussed the discrepancy between these studies and our work (line 308-318, page 12 in this revision).

In addition, as shown in supplementary Fig. 2, our data showed that DFO did not only affect curcumin-induced increase in intracellular ferrous ions, but also decrease the control level of ferrous ions (the second column).

Question #4/Response #4(2): The authors correct the unit of ferrous ion μg/μg protein to nmol/μg protein. According to the abcam’s protocol (ab83366, abcam) the authors used, rat liver lysate contains approximately 0.1 nmoles Fe2+/mg tissue, which is approximately 10,000 times less compared the amount of Fe2+ the authors reported (10 µmoles/µg protein).

Response: In this study, the unit of ferrous was (nmol/μg protein) × 103 in Fig. 2 and 3, which is equivalent to nmol/mg protein.

Question #5/Response #5: Erastin at 10 nM cannot cause ferroptotic cell death in cultured cells. Most cancer cells are required 5-10 µM erastin to cause ferroptosis.

Response: We are very sorry for our carelessness in writing. After careful checking, we verified that the dose of erastin used in this experiment was 10 µM. Special thank you very much for your careful review and we express our sincere apologies for this error. We corrected this mistake in this revision (line 90, page 4 and line 501, page 18 in this revision).

---

## [Decision Letter · Decision Letter 2]

12 Nov 2021

PONE-D-21-20659R2Curcumin suppresses tumorigenesis by ferroptosis in breast cancerPLOS ONE

Dear Dr. Guan,

Thank you for submitting your manuscript to PLOS ONE. After careful consideration, we feel that it has merit but does not fully meet PLOS ONE’s publication criteria as it currently stands. Therefore, we invite you to submit a revised version of the manuscript that addresses the points raised during the review process.

Academic editor: Please, check the results mentioned by the Reviewer.

We look forward to receiving your revised manuscript.

Kind regards,

Irina V. Balalaeva, PhD

Academic Editor

PLOS ONE

Journal Requirements:

Reviewers' comments:

Reviewer's Responses to Questions

**Comments to the Author**

1. If the authors have adequately addressed your comments raised in a previous round of review and you feel that this manuscript is now acceptable for publication, you may indicate that here to bypass the “Comments to the Author” section, enter your conflict of interest statement in the “Confidential to Editor” section, and submit your "Accept" recommendation.

Reviewer #1: All comments have been addressed

2. Is the manuscript technically sound, and do the data support the conclusions?

Reviewer #1: Yes

3. Has the statistical analysis been performed appropriately and rigorously? 

Reviewer #1: Yes

4. Have the authors made all data underlying the findings in their manuscript fully available?

Reviewer #1: Yes

5. Is the manuscript presented in an intelligible fashion and written in standard English?

Reviewer #1: No

6. Review Comments to the Author

Reviewer #1: The author’s response: In addition, as shown in supplementary Fig. 2, our data showed that DFO did not only affect curcumin-induced increase in intracellular ferrous ions, but also decrease the control level of ferrous ions (the second column).

The authors did not respond my concern as follows:

DFO only affected curcumin-induced increase in intracellular ferrous ions (Figure S2A, the 3rd column vs 4th column) but did not affect the control level of ferrous ions (Figure S2A, the first column vs second column). The results is strange.

The English writing standard needs to be improved further.

7. PLOS authors have the option to publish the peer review history of their article (what does this mean?). If published, this will include your full peer review and any attached files.

Reviewer #1: No

---

## [Author Response · Author response to Decision Letter 2]

24 Nov 2021

Reviewer #1:

Question #1: The authors did not respond my concern as follows:

DFO only affected curcumin-induced increase in intracellular ferrous ions (Figure S2A, the 3rd column vs 4th column) but did not affect the control level of ferrous ions (Figure S2A, the first column vs second column). The results are strange.

Response #1: We retest the intracellular ferrous ions in MDA-MB-453 and MCF-7 cells with or without the presence of DFO. As shown in Figure 1 in this response, no obvious difference was observed between the intracellular ion level in of MDA-MB-453 and MCF-7 cells with or without the presence of 50 μM deferoxamine, which was also found in recent published researches [1-3]. Although deferoxamine mesylate was an iron chelator that bound free iron in a stable complex, we speculated that it might not affect the relatively low normal ferrous ions level in some cells but effectively reverse the iron accumulation induced by ferroptosis inducer.

Question #2: The English writing standard needs to be improved further.

Response #2: As suggested, we improved the quality of our language by an English language editing service of Spandidos Publications. Based on the editing report, we revised our manuscript and also uploaded the certificate with this revision.

References in this response:

1. Tu H, Zhou YJ, Tang LJ, Xiong XM, Zhang XJ, Ali Sheikh MS, et al. Combination of ponatinib with deferoxamine synergistically mitigates ischemic heart injury via simultaneous prevention of necroptosis and ferroptosis. Eur J Pharmacol. 2021;898:173999. Epub 2021/03/07. doi: 10.1016/j.ejphar.2021.173999. PubMed PMID: 33675785.

2. Yang J, Zhou Y, Xie S, Wang J, Li Z, Chen L, et al. Metformin induces Ferroptosis by inhibiting UFMylation of SLC7A11 in breast cancer. J Exp Clin Cancer Res. 2021;40(1):206. Epub 2021/06/25. doi: 10.1186/s13046-021-02012-7. PubMed PMID: 34162423; PubMed Central PMCID: PMCPMC8223374.

3. Song Z, Xiang X, Li J, Deng J, Fang Z, Zhang L, et al. Ruscogenin induces ferroptosis in pancreatic cancer cells. Oncol Rep. 2020;43(2):516-24. Epub 2020/01/03. doi: 10.3892/or.2019.7425. PubMed PMID: 31894321; PubMed Central PMCID: PMCPMC6967081.

---

## [Editor Report · Decision Letter 3]

1 Dec 2021

Curcumin suppresses tumorigenesis by ferroptosis in breast cancer

PONE-D-21-20659R3

Dear Dr. Guan,

We’re pleased to inform you that your manuscript has been judged scientifically suitable for publication and will be formally accepted for publication once it meets all outstanding technical requirements.

Kind regards,

Irina V. Balalaeva, PhD

Academic Editor

PLOS ONE
---

## [Editor Report · Acceptance letter]

7 Jan 2022

PONE-D-21-20659R3 

Curcumin suppresses tumorigenesis by ferroptosis in breast cancer 

Dear Dr. Guan:

I'm pleased to inform you that your manuscript has been deemed suitable for publication in PLOS ONE. Congratulations! Your manuscript is now with our production department. 

Kind regards, 

on behalf of

Dr. Irina V. Balalaeva 

Academic Editor

PLOS ONE